# Identification of a DEAD-box RNA Helicase BnRH6 Reveals Its Involvement in Salt Stress Response in Rapeseed (*Brassica napus*)

**DOI:** 10.3390/ijms24010002

**Published:** 2022-12-20

**Authors:** Xianduo Zhang, Jianbo Song, Liping Wang, Zhi Min Yang, Di Sun

**Affiliations:** 1Department of Biochemistry and Molecular Biology, College of Life Sciences, Nanjing Agricultural University, Nanjing 210095, China; 2Department of Biochemistry and Molecular Biology, College of Science, Jiangxi Agricultural University, Nanchang 330045, China

**Keywords:** *Brassica napus*, salt stress, RNA helicase, transcriptome

## Abstract

Rapeseed (*Brassica napus*) is one of the most important vegetable oil crops worldwide. Abiotic stresses such as salinity are great challenges for its growth and productivity. DEAD-box RNA helicase 6 (RH6) is a subfamily member of superfamily 2 (SF2), which plays crucial roles in plant growth and development. However, no report is available on RH6 in regulating plant abiotic stress response. This study investigated the function and regulatory mechanism for *BnRH6*. BnRH6 was targeted to the nucleus and cytoplasmic processing body (P-body), constitutively expressed throughout the lifespan, and induced by salt stress. Transgenic overexpressing *BnRH6* in *Brassica* and *Arabidopsis* displayed salt hypersensitivity, manifested by lagging seed germination (decreased to 55–85% of wild-type), growth stunt, leaf chlorosis, oxidative stress, and over-accumulation of Na ions with the K^+^/Na^+^ ratio being decreased by 18.3–28.6%. Given the undesirable quality of knockout *Brassica* plants, we utilized an *Arabidopsis* T-DNA insertion mutant *rh6-1* to investigate downstream genes by transcriptomics. We constructed four libraries with three biological replicates to investigate global downstream genes by RNA sequencing. Genome-wide analysis of differentially expressed genes (DEGs) (2-fold, *p* < 0.05) showed that 41 genes were upregulated and 66 genes were downregulated in *rh6-1* relative to wild-type under salt stress. Most of them are well-identified and involved in transcription factors, ABA-responsive genes, and detoxified components or antioxidants. Our research suggests that *BnRH6* can regulate a group of salt-tolerance genes to negatively promote *Brassica* adaptation to salt stress.

## 1. Introduction

Agroecosystems are complex systems full of multi-environmental challenges, including biotic and abiotic stresses constantly encountered by sessile plants. Salinity stress is one of the most severe abiotic stresses for most terrestrial plants [1]. Plants grown in excess salt environments suffer from osmotic stress and ionic toxicity that disturb numerous metabolisms, antioxidative systems, and consequent growth and development [2,3,4]. To deal with the adverse environmental challenge, plants have developed diverse mechanisms, including the altered expression of massive stress responsive and tolerant genes [5,6]. The well-known transcription factors and kinases in signaling cascades, for example, are good cases in point [5,6].

RNA helicases are a class of ATPases functioning in RNA duplex unwinding and ribonucleoprotein (RNP) structure remodeling [7]. Based on the sequence, the RNA helicase can be classified into six superfamilies (SFs) [8]. SF2 is the major superfamily. Based on the amino acid sequence of conserved motif II, SF2 can be further divided into three subfamilies: (1) DEAD (Asp-Glu-Ala-Asp)-box; (2) DEAH (Asp-Glu-Ala-His)-box; and (3) DExD/H-box [9]. DEAD-box RNA helicases are the largest subgroup and participate in almost all RNA biological processes, such as RNA transcription, RNA transport, mRNA splicing, RNA silencing, and RNA decay [10]. They are also actively involved in plant growth, development, and plant–environment interactions [11,12]. To date, a bunch of DEAD-box RNA helicase genes has been shown to play roles in plant abiotic stress response regulation. For example, low expression of osmotically responsive genes 4 (*LOS4*, a DEAD-box RNA helicase gene) is required for the low temperature tolerance by promoting mRNA export in *Arabidopsis* [13]. *OsBIRH1* is a DEAD-box RNA helicase gene in rice; heterologous overexpression of *OsBIRH1* in *Arabidopsis* increased oxidative stress tolerance by elevating the expression of antioxidative genes [14]. In *Arabidopsis*, RNA helicase 3 (AtRH3) confers salt stress tolerance by splicing mRNA in chloroplast under salt stress [15]. Nevertheless, most DEAD-box RNA helicase members remain to be investigated, particularly the ones responding to abiotic stresses. 

*Brassica napus* (*Brassica*) is an amphidiploid *Brassica rapa* and *Brassica oleracea* [16]. *Brassica* and *Arabidopsis thaliana* (*Arabidopsis*) belong to the same cruciferous family. As one of the main resources of vegetable oil, *Brassica* is cultivated on upland over the world and is vulnerable to a variety of adverse environmental stresses such as drought, salinity, and unfavorable temperature. Identifying the gene function related to abiotic stress tolerance would contribute to breeding varieties that ensure the vegetable plants are more productive under adverse environments. We previously isolated two DEAD-box RNA helicase genes in *Brassica* under environmental stress, both are homologs of DDX6 (DEAD-box RNA helicase 6) in animals and RH6 (RNA helicase 6) in *Arabidopsis*. Recent studies show that DDX6 and AtRH6 are involved in RNA turnover and miRNA biogenesis [17,18,19], but whether they function in plant salt stress response remains unknown. Here, we identified an uncharacterized locus encoding a DDX6 homolog gene *BnRH6* in *Brassica*. We show that BnRH6 is localized to the nucleus and processing-body (P-body) in the cytoplasm, and can be upregulated under salt stress at the transcription level. Transgenic overexpressing *BnRH6* in *Brassica* and *Arabidopsis* revealed hypersensitivity to salinity, suggesting a negative role of *BnRH6* played in tolerance to salt stress. We took advantage of an AtRH6 T-DNA insertion mutant line *rh6-1* to profile the genome-wide transcripts under salt stress and found significant changes in the expression of many salt-tolerant genes, including signaling components (phytohormones and transcription factors), and proteins or enzymes for metabolism and antioxidation. Our work supports the important role of *BnRH6* in the negative regulation of the *Brassica* adaptation salinity environment and unveils its downstream genes with mechanistic pathways leading to the plant response to salt stress. 

## 2. Results

### 2.1. BnRH6 Is Transcriptionally Upregulated under Salt Stress

Two homologues, *BnRH6-1* (*DEAD-box RNA helicase 24*, BnaA04g26450D, named *BnRH6* in this article) and *BnRH6*-*2* (*DEAD-box RNA helicase 100*, BnaC04g50480D), were identified in *Brassica* (Appendix A) [16,20]. Alignment of amino acids of *BnRH6* revealed that there are typical DEAD-box RNA helicase domains with high similarity within *Brassica* and *Arabidopsis* (Appendix A). RH6s have two homologs, RH8s and RH12s (Appendix A), which is very similar to those of the *Arabidopsis* species, indicating that *BnRH6s* are highly conserved in Brassicaceae plants including *Brassica* and *Arabidopsis*. 

*BnRH6* was transcriptionally expressed in various tissues or organs throughout the lifespan. In young seedlings, *BnRH6* is dominantly expressed in euphylla and hypocotyl and moderately in other tissues such as root and cotyledon (Figure 1A). At early developmental stages, *BnRH6* is evenly expressed in all tissues. During flowering and seed developing stage, *BnRH6* showed a higher expression in stems, flowers, and siliques but a relatively lower expression in leaves (Figure 1A). To examine whether *BnRH6* was induced by salt stress, two-week-old seedlings were treated with NaCl at 100, 200, and 300 mM. When plants were exposed to 100 mM NaCl for 6 h, the *BnRH6* transcript levels in roots and shoots were increased by 2.5- and 2.6-fold higher than those of the control, respectively (Figure 1B). At 200 mM NaCl, *BnRH6* expression declined but remained at a significantly higher level over the control. These results indicated that *BnRH6* is transcriptionally induced by salt stress.

### 2.2. BnRH6 Is Localized to the Nucleus and Cytoplasm

We further examined the subcellular localization of the BnRH6 protein. The *BnRH6* coding sequence was amplified and fused to the green fluorescent protein (GFP) by linking the N- or C-terminal of *BnRH6*. The fusion and GFP alone were further ligated to a vector with a 35S promoter to drive its expression. In the meantime, two fluorescent marker proteins including MADS3 targeted to the nucleus and AtDCP1 to the cytoplasm were obtained. The vectors were transferred into the leaf cells of tobacco. GFP alone served as a technical control (Figure 1C). The BnRH6-GFP fusions with green signals and marker proteins with red signals were dominantly expressed in the same regions of the nucleus and cytoplasm. The overlapping colors merged to create yellow ones (Figure 1D,E), indicating that BnRH6-GFP can be targeted on both sides. Since AtDCP1 was reported as a P-body marker [21], there is a likelihood that BnRH6 is associated with P-body. 

### 2.3. BnRH6 Overexpressed Brassica and Arabidopsis Display Salt Sensitivity

To evaluate the role of *BnRH6* under salinity stress, we generated *BnRH6*-overexpressing *Brassica* (*BnOE*) and *Arabidopsis* (*AtOE*) lines (Appendix A). Three transformants of each *BnOE* and *AtOE* plant (T3 Homozygotes) were randomly selected, genotyped at DNA levels (Appendix A) and determined at transcriptional levels (Appendix A). No obvious differences in phenotypes were observed between wild-type (WT) and *BnOE* lines without NaCl treatment (Figure 2A,B). When exposed to 100, 125, and 150 mM NaCl, however, the *BnOE* plants were more sensitive to salt stress than WT (Figure 2A,B). Consistently, the seed germination ratio between WT and *BnOE* plants remained unchanged in the absence of NaCl, while following a three-day exposure to 100, 125, and 150 mM NaCl, the seed germination ratio of *BnOE* lines decreased to 55–85% of the WT, respectively (Figure 2C). Similar responses were observed for the cotyledon greening rate (Figure 2D). Nearly all WT and *BnOE* seeds exhibited cotyledon greening on the 3rd day without NaCl, while more leaf chlorosis symptoms were observed in *BnOE* plants than in WT under NaCl exposure (Figure 2D). Furthermore, the growth of root and shoot and fresh biomass of *BnOE* seedlings was more severely inhibited under salt stress compared with that of WT (Figure 2E–G). With regard to *AtOE*, similar responses concerning seed germination, cotyledon greening, root elongation, and biomass were observed (Appendix A). These results suggest that BnRH6-overexpression repressed seed germination under salt stress.

We conducted an additional study with 10-day-old *BnOE* and its WT plants treated with NaCl. While both varieties showed no evident growth difference under normal conditions, the *BnOE* plants displayed compromised growth status relative to WT (Figure 3A,B). The root length and shoot height of *BnOE* plants were reduced by 8.14–18.79% and 11.97–19.34% under NaCl treatment, and the dry weight of root and shoot of *BnOE* seedlings dropped by 14.47–23.47% and 15.05–29.69%, respectively (Figure 3C–F). Likewise, the *AtOE* lines displayed more growth defects in root elongation and shoot growth than WT plants (Appendix A). These results pointed out that overexpression of *BnRH6* could make the plant vulnerable to salt stress. 

Chlorophyll and malondialdehyde (MDA) concentrations are typical physiological indicators reflecting plant growth status and stress-induced generation of peroxides under salinity stress, respectively [22,23]. As shown in (Figure 4A), when *BnOE* and WT were exposed to NaCl, the leaf of *BnOE* lines became much more chlorotic, particularly at higher degrees of salt stress (Figure 4A). The leaf phenotype in *AtOE* plants with salt treatment showed a similar chlorosis symptom (Figure 4B). Assessments of chlorophyll revealed that the chlorophyll concentrations in *BnOE* and *AtOE* plants significantly decreased as compared with those of wild-type (Figure 4C,D). Meanwhile, measurements of MDA showed that the MDA concentrations were significantly increased in the transgenic plants (Figure 4E,F). These results suggest that *BnRH6* overexpression could also impair the physiological response under salt stress.

### 2.4. BnRH6 Overexpression Promotes Na^+^ Accumulation in Plants

To identify whether the *BnRH6*-mediated compromised growth and physiological response is associated with the accumulation of Na^+^ in plants, concentrations of Na ions in roots and shoots were separately determined in the transgenic and wild-type plants. An ICP-MS analysis showed that Na^+^ levels were significantly higher in both roots and shoots of the *BnOE* than in WT plants, especially at the high level of NaCl (200 mM) (Figure 5A,B). The enhanced Na^+^ levels were also detected in roots and shoots of the *AtOE* lines, where the Na^+^ concentrations increased by 61.5–73.1% and 14.3–33.3% compared with wild-type, respectively (Appendix A). Since the ratio of K^+^/Na^+^ is commonly used as one of the important physiological responses for plant tolerance to salt stress [24], we simultaneously measured the K ions in the plants. The K^+^ concentrations were comparable between the overexpression lines and wild-type (Figure 5C,D and Appendix A). However, calculations of the K^+^/Na^+^ ratio showed that the ratio declined in the *BnOE* and *AtOE* plants compared with the wild-type plants. The significant differences between the transgenic plants and wild-type occurred at 150–200 mM of NaCl (Figure 5E–G and Appendix A). In the shoot of the *BnOE* plants, the K^+^/Na^+^ ratio decreased by 18.3–28.6% compared with wild-type (Figure 5G). The results suggest that overexpression of *BnRH6* results in the over-accumulation of Na ions.

### 2.5. Suppression of RH6 Enhances Plant Tolerance to Salt Stress by Upregulating the Expression of Salt Stress-Tolerant Genes in Arabidopsis

To figure out the mechanisms or pathways responsible for plant tolerance to salinity regulated by RH6, we performed a genome-wide identification of transcriptomes by RNA sequencing. Due to the lack of desirable knockout lines of *Brassica*, we took an advantage of an *Arabidopsis* knockout mutant *rh6*-*1*. The *rh6*-*1* mutant with a T-DNA insertion in an exon near 5′-UTR was identified by PCR (Appendix A). We applied the *atrh6-1* for the study because both BnRH6 and AtRH6 are highly conserved, by which the amino acid sequences between BnRH6 and AtRH6 share an 85.72% similarity, and the key domains, DEAD domain (ATPase domain including Q, I, Ia, Ib, II and III motif) and HELIC domain (RNA binding domain including III, IV and VI motif) between BnRH6 and AtRH6 share similarities of 98.98% and 97.56%, respectively (Appendix A). Furthermore, the *rh6-1* mutant displays an enhanced salt-tolerance phenotype with better growth (longer root length and greater biomass) compared with the wild-type (Col-0) (Appendix A), confirming that both AtRH6 and BnRH6 should be negative regulators of plant response to salt stress. 

Four RNA libraries regarding *rh6*-*1* and WT connecting -Na or +Na were constructed in terms of WT(−Na), WT(+Na), *rh6-1*(−Na), and *rh6-1*(+Na) with three biological replicates (4 × 3 samples) and sequenced by high-throughput RNA-sequencing using Illumina technology. The quality of the sequencing datasets was statistically evaluated, and it turned out to be reliable (Appendix A and Appendix A; Appendix A). To validate the expression pattern of the genes determined by RNA-seq, a total of 21 genes were selected for qRT-PCR validation. Expression of all genes showed a pattern identical to the RNA-seq (Appendix A). 

The differentially expressed genes (DEGs) (>2-fold change, *p*< 0.05) from the group pairs were identified, and the number of each pair was presented (Figure 6A,B, Appendix A). The total number of DEGs (including up and down genes) between the two datasets WT(+Na)/WT(−Na) and *rh6-1*(+Na)/*rh6*-*1*(−Na) were comparable, while the number of DEGs between *rh6-1*(−Na)/WT(−Na) and *rh6-1*(+Na)/WT(+Na) were drastically changed (Figure 6A). Venn diagram analysis revealed that 342 DEGs were specifically upregulated and 281 DEGs were downregulated in *rh6-1*(+Na)/*rh6-1*(−Na), whereas 433 DEGs were specifically induced and 213 DEGs were repressed in WT(+Na)/WT(−Na) (Figure 6B). Further cross-analyses between *rh6-1* and WT showed that 40 DEGs were specifically induced and 62 DEGs were suppressed in the *rh6-1*(+Na)/WT(+Na) dataset, while 27 DEGs were specifically upregulated and 44 DEGs were repressed in the *rh6-1*(−Na)/WT(−Na) pair (Figure 6C). Notably, compared to the minus sodium (−Na), treatment with plus sodium (+Na) induced 1.48-fold (40/27) more DEGs in *rh6-1*(+Na)/WT(+Na) than in *rh6-1*(−Na)/WT(−Na) datasets; on the other hand, the *rh6-1*(+Na)/WT(+Na) group contained 1.41-fold (62/44) more downregulated DEGs in *rh6-1*(+Na)/WT(+Na) than in *rh6-1*(−Na)/WT(−Na) (Figure 6C). These results point out that the mutation of *RH6* would tend to alter the pattern of more gene expressions under salt stress. 

We then performed Gene Ontology (GO) analysis and placed the DEGs into some functional categories related to salt stress response. While under the −Na condition, many DEGs in the *rh6-1* mutant filled into the development of the biological process, and the DEGs of *rh6-1*(+Na) vs. *rh6*-*1*(−Na) were found with enrichment in the categories such as stimuli, chemical stress, and hormone response (Appendix A). It was important to find that in the dataset of *rh6-1*(+Na)/WT(+Na), the DEGs were also classified into stress responses and different kinds of metabolic processes (Appendix A). These results indicate that under salt stress the *rh6-1* function most likely would shift from the developmental state to the stress responsive model by post-transcriptional regulation. The Kyoto Encyclopedia of Genes and Genomes (KEGG) refers to the genes categorized into specific pathways. The DEGs in the dataset *rh6-1*(−Na)/WT(−Na) were concentrated in the pathways of RNA degradation, MAPK and hormone-signaling pathway, metabolite biosynthesis, and plant–pathogen interaction (Appendix A). Similar results were found in the *rh6-1*(+Na)/WT(+Na) and *rh6-1*(+Na)/*rh6-1*(−Na) datasets, in which some DEGs participated in the stress signal, metabolite biogenesis, and RNA metabolism pathways (Appendix A). These results suggest that mutation of *RH6* led to many stress-responsive genes being involved in the salt resilience process. 

We further specified the DEGs in the datasets that were regulated in the *rh6-1* mutant line under salt exposure. Four heatmaps were drawn based on the expression differences of the DEGs between the groups. In the datasets of WT(+Na)/WT(−Na) and *rh6-1*(+Na)/*rh6*-*1*(−Na), there were 34 salt-responsive genes in response to salt stress. Of these, 33 genes were upregulated more in *rh6-1* compared with WT (Figure 6D, Appendix A). Of those, six genes encoding transcription factors including *MYB112*, *NAC019*, *NAC032*, *NAC071*, *PLATZ*, and *CBF4* were detected; four genes *NCED9*, *ABR*, *ABI1*, and *HAIs* (ABA pathway) were associated with phytohormone synthesis or signals; one gene encoding kinase involved MAPK cascades (*MAPKKK18*); two genes encoded antioxidative enzymes (*GSTUs*); one gene encoded an ion exchange protein (*CAX3*); and the rest of them were stress-related genes,including Late Embryogenesis-Abundant proteins (*LEAs*) (*LEA14*, *LEA7*, *AT2G18340*, *LEA18*, *ABR*, *AT3G53040, COR47* and *LEA4*-5) and *CYP450s* (*CYP707A1*, *CYP94B3* and *CYP81G1*) (Figure 6D, Appendix A). One *Seed Storage Protein* (*SSP*) gene was downregulated more in *rh6-1* relative to WT (Figure 6D, Appendix A). In particular, 39 salt-related genes were uniquely upregulated in *rh6-1*(+Na)/*rh6*-*1*(−Na) but not in WT(+Na)/WT(−Na) (Figure 6E, Appendix A), including 14 transcription factor genes (2 *HSFAs*, 2 *PLATZs*, 3 *NACs*, 2 *WRKYs* and 5 *MYBs*), an ABA responsive gene (*AHG1*), four *CYP450s*, seven encoding antioxidative enzymes (*TH8*, *TRX5*, *AT1G03020*, *AT3G11773*, *AT5G52410*, *AT2G22420* and *AT2G18150*), one gene encoding enzyme in proline synthesis (*P5CS2*), one gene encoding ion channel (*SLAH3*), two *LEA.* Most of them have been reported to involve salt or abiotic stress responses. Two DEGs involved in salt stress response were also found in datasets of *rh6-1*(+Na)/WT(+Na) and *rh6-1* (−Na)/WT(−Na) (Figure 6F, Appendix A).

## 3. Discussion

### 3.1. Expression of BnRH6 Compromises Growth of Plants under Salt Stress

In this study, we report that BnRH6 is a new regulator in salt stress response by downregulating salt-tolerant genes. *BnRH6* was transcriptionally expressed in various tissues or organs all lifelong, indicating a potential role in all stages (Figure 1A). This wide expression pattern is in agreement with its RH6 in *Arabidopsis* and other RNA helicases [25,26,27,28]. Moreover, BnRH6 was localized in the nucleus and cytoplasm (most likely in P-body) (Figure 1C–E), suggesting potential functions on both sides. This is reminiscent of RH6 in *Arabidopsis*, which plays roles in both the nucleus and cytoplasm [19,25]. There is a likelihood that BnRH6 is associated with P-body, similar to the report on AtRH6 [25]. 

The transcript levels were increased drastically in the shoot and root of *Brassica* with NaCl exposure, implying a possible role in salt stress response (Figure 1B). Several reports have indicated that the expression of RNA helicases correlated with salt stress response can be regulated by salinity stress in multiple species including *Arabidopsis*, rice, cabbage, tomato, barley, and so on [28,29,30,31,32]. However, not all salt stress response-related RNA helicases can be induced at the transcript level. For instance, *Arabidopsis AtRH17* is a positive regulator to plant salt stress tolerance, whereas the *AtRH17* transcript level was not affected under salinity stress [33]. Furthermore, a certain number of RHs participate in more than one kind of abiotic stress [28,29,30,32,34]. Whether *BnRH6* expression can be altered by other stresses needs to be further investigated.

Genetic and functional investigation revealed that *BnRH6* overexpression in both *Brassica* and *Arabidopsis* led to plant sensitivity to salt stress, in terms of a decreased germination ratio and cotyledon greening rate, weakened growing status, and more sensitive physiological reactions. This scenario is consistent with our RNA-seq data. *BGLU31* and *BGLU32* encode β-glucosidases as positive factors for seed germination [35] and were found to be upregulated in *rh6-1* under salt stress (Figure 6D). Notably, the gene *SSP* negatively mediating seed germination was found to be downregulated in *rh6-1* plants (Figure 6D,E) [36]. Several classes of growth-regulating genes, such as *ELIP1* and *ELIP2* encoding chlorophyll a/b binding family protein genes and *ARR18* encoding response regulator genes, were all upregulated in the *rh6-1* mutant lines (Figure 6E). Thus, *BnRH6* affects the salt stress response likely via indirect control of the expression pattern of those genes. 

Overload of NaCl into plants initially triggers osmotic stress and later ion imbalance and toxicity [37,38]. Proline is an osmotic regulator induced by salt stress. One gene encoding an enzyme (P5CS2) in the proline synthesis was specifically upregulated in salt-exposed *rh6-1* plants (Figure 6E). Our data showed that the Na^+^ accumulation was increased in shoots and roots of *BnRH6* overexpression lines. Even though the K^+^ accumulation was not altered in overexpression lines, a relatively lower K^+^/Na^+^ ratio was generated (Figure 5F,G). The Na^+^ accumulation and K^+^/Na^+^ ratio was barely discussed in RNA helicase gene mutants or overexpression plants involved in the salt-stress response. Similar changes have been evident in other salt-stress regulator mutants or overexpression lines. For instance, *AtDIF1* overexpression *Arabidopsis* conferred plant tolerance to salt stress by reducing the content of Na^+^ and loss of K^+^ [39]. SERF1 is a positive regulator for salt stress tolerance in rice. Loss-of-function or the knock-down mutant of *SERF1* showed an increased Na^+^ accumulation and a higher Na^+^/K^+^ ratio [40]. 

Our RNA seq data showed that one ion channel gene, *CAX3* was induced in *rh6-1* under salt stress (Figure 6D,E). A previous study showed that *CAX3* can be induced by Na^+^ treatment. CAX3 is a Ca^2+/^H^+^ vacuolar antiporter for plant tolerance to salt under Na^+^ stress. The *cax3* mutant confers decreased P-ATPase activity, which further decreased Na^+^ efflux [41]. On the contrary, an enhanced expression of CAX3 may promote Na^+^ efflux and benefit the maintenance of a high K^+^/Na^+^ ratio. Apart from the notorious detriment caused by Na^+^, the accompanied Cl^−^ poisoning is also fatal [42,43]. SLAH3 is a Cl^−^ efflux transporter [44]. Transcriptional *SLAH3* was downregulated by NaCl treatment [45]. An increased expression of *SLAH3* in *rh6-1* under NaCl exposure was recovered from our RNA-seq data, which may decrease the Cl^−^ accumulation. These results suggest that *CAX3* and *SLAH3* could play critical roles in ion homeostasis under salt stress.

### 3.2. Mutation of rh6-1 Confers Plant Salt Tolerance by Regulating Salt-Resilient Genes

To identify the downstream genes and possible regulation pathways governed by RH6 under salt stress, we performed a genome-wide analysis of transcriptome across the *rh6-1* and Col-0 plants. While a large number of DEGs were expressed in the *rh6-1* and wild-type plants under salt stress, there were many DEGs across the two cultivars exposed to NaCl (>2.0-fold change, *p* < 0.05). The largest group of specific DEGs was associated with genes encoding transcription factors (TFs), whose expression patterns were altered in the *rh6-1* mutant. Nine MYB genes were found to be upregulated in *rh6-1* mutant after salt treatment. MYB proteins play a fundamental role in numerous aspects of plant growth, development, and stress responses [46,47]. *MYB80*, *MYB113,* and *MYB114* were upregulated in *rh6-1* plants relative to WT only under salt stress (Figure 6G, Appendix A). MYB113 and MYB114 have been reported to promote anthocyanin synthesis in *Arabidopsis* and pear for defense against abiotic and biotic stresses [48,49]. *MYB112* was strongly expressed in *rh6-1* under salt stress (Figure 6D) and is also required for anthocyanin accumulation under salinity and high light stress [50]. Several other gene members including *MYB37*, *MYB62*, *MYB90*, and *MYB120* were also induced in *rh6-1* under NaCl stress (Figure 6E). Some of them are also involved in abiotic stress responses [51,52,53]. 

Several *NAC* (*NAM*, *ATAF1/2* and *CUC2*) genes were detected in *rh6-1* plants under salt stress including *NAC019*, *NAC032* and *NAC071* (Figure 6D) and *NAC044*, *NAC061,* and *NAC096* (Figure 6E), and most of them have been reported to involve salt and other abiotic stress response [54,55]. *Arabidopsis* harboring an overexpressed *GmNAC019* increased drought and salinity tolerance [56]. *NAC061* is induced by AtHSFA7b (a heat shock transcription factor in *Arabidopsis*) and positively regulates salt tolerance [57]. In addition to MYB and NAC, several other TF family genes such as *WRKY45* and *CBF4* identified in the study are also reported to regulate plant response to salt stress [58,59]. Whether BnRH6 directly or indirectly interacts with TFs remains to be investigated.

Plant hormones are the major regulators of abiotic stress responses [60]. ABA has been long considered to be a crucial driver of plant abiotic stress, especially salinity and dehydration stress responses [61,62]. In this study, seven ABA signaling pathway genes for NCED9, ABI1, HAIs, AHG1, MAPKKK18, and CYP707A1 were identified as salt-responsive DEGs. *Arabidopsis* NCED9 (9-cis-epoxycarotenoid dioxygenase 9) is essential for ABA biosynthesis [63,64]. Expression of *NCED9* is induced by salt, drought, and cold in *Brassica oleracea* [65]. Several DEGs encode ABI1, AHG1, HAI1, HAI2, and HAI3, and these proteins are phosphatases 2Cs (PP2Cs) responsible for ABA signal transduction [66,67]. CYP707A1 and MAPKKK18 were also involved in ABA-related stress responses [68,69,70]. This is reminiscent of some recent reports that other DEAD-box RNA helicases such as BnRH37 improved abiotic stresses of plants by inducing the expression of ABA biosynthesis-related genes or ABA-dependent stress-response genes [33,71,72], but no report was published for BnRH6.

It is well known that salinity stress is usually accompanied by oxidative stress [73,74]. Under salt stress, some anti-oxidative enzymes such as glutaredoxins and thioredoxins actively participate in the removal of salt stress-induced ROS and toxicity [75]. Five genes encoding ROS scavengers were upregulated in *rh6-1* plants. For example, *GRXS4*/*ROXY13* and *TH8* were transcriptionally upregulated in *rh6-1* mutants under NaCl stress. GRXS4 belongs to the CC-type glutaredoxin (ROXY) protein family, and was reported to be induced by salinity and drought stresses [76,77]. Both *TRX5* (*thioredoxin H-type 5*) and *TH8* (*thioredoxin H-type 8*) encode thioredoxins and can be induced by multiple biotic and abiotic stresses [78]. GSTU4 and GSTU6 are tau class glutathione S-transferases; GSTU6 was reported to contribute to Cd stress tolerance by regulating intracellular ROS homeostasis in rice [79]. Overexpressing *GmGSTU4* displays salt tolerance in tobacco through peroxidase activity and detoxification mechanisms [80,81]. In addition, genes indirectly participating in ROS scavenging were screened. For example, *PtrLEA7*, a gene encoding a Late embryogenesis-abundant (LEA) protein, can be slightly induced by salt stress; overexpressing *PtrLEA7* in tobacco and *Poncirus trifoliata* positively regulates drought tolerance through enhancing antioxidant capacity [82]. Similarly, an RNA helicase VviDEADRH25 in grapevine plays a role as a negative regulator to drought stress tolerance. Overexpressing *VviDEADRH25* in *Arabidopsis* decreased the expression of genes encoding ROS scavenging enzymes [83]. These results suggest that *rh6-1*-mediated plant tolerance to salt stress would be likely through these homolog genes. 

Apart from the subsets above, we also detected some other protein family genes involved in the RH6-mediated pathways under salt stress. LEA proteins belong to a large hydrophilic protein family with regulatory roles in plant stress responses [84]. We found that *LEA4*-*5*, *LEA7*, *LEA14*, and *LEA18* were significantly upregulated in *rh6* under salt stress. LEA14 was previously reported to confer salt stress tolerance by stabilizing an E3 ligase in *Arabidopsis* [85]. *LEA3* was induced only in *rh6* but not in WT under Na stress, suggesting that *LEA3* was most likely involved in the salt stress response. Supportive evidence comes from the rice *LEA3* because the *OsLEA3*-overexpression line endowed drought resistance in the plant [86]

These results suggest that the RH6 is likely a master regulator of RNA biosynthesis and metabolism under salt stress. Based on those results, we proposed a model of salt stress response in WT and BnOE *Brassica* (Figure 7). Our research found a new function of BnRH6, and it may be a promising molecular tool for engineering salt-tolerant *Brassica* and enriching *Brassica* varieties.

## 4. Materials and Methods

### 4.1. Plant Growth and Salt Treatments

Seeds of *Brassica* (genotype: Westar) were surface-sterilized in 75% ethanol alcohol for 5 min and 10% NaClO for 15 min with gentle shakes, followed by thoroughly rinse with sterile water. The clean seeds were placed on a plastic net floating on the 1/2 Hoagland solution. After germination at 25 °C in the dark for two days, seeds were grown under the condition of 22 °C, 200 μmol m^−2^s^−1^ light intensity and 14 h/10 h(light/dark) photoperiod [87]. Plant tissues at different stages were harvested for *BnRH6* expression pattern analysis. Two-week-old *Brassica* seedlings were exposed to 1/2 Hoagland solution (pH = 5.8) with different NaCl concentrations (0, 100, 200, and 300 mM) for 6 h based on previous reports [88]. Shoots and roots were harvested separately for expression analysis.

Seeds of *Arabidopsis* (Col-0) and a *rh6-1* T-DNA insertion mutant line (SALK_ 205997C, Columbia background, obtained from the *Arabidopsis* Biological Resource Center) were sterilized and sowed on ½ Murashige and Skoog (MS) medium supplemented with 3% sucrose and 0.9% agar. After two-day vernalization at 4 °C in darkness, seeds were grown in a growth chamber with 22 °C, 150 μmol m^−2^s^−1^ light intensity and 16/8 h (light/dark) photoperiod. 

### 4.2. Vector Construction and Transgenic Plants

Full-length CDS sequence of *BnRH6* (*Bn04g26450D*) was cloned using specific primers (Appendix A) and inserted into the pCAMBIA1300 vector with CaMV35S as the promoter [89]. The sequence-confirmed construct was transformed into *Agrobacterium tumefaciens* strain EHA105 following the standard method of transformation [90]. The Agrobacterium-mediated plant transformation was performed with *Brassica* hypocotyl sections [87] for *Brassica* plants and floral dip method for *Arabidopsis* plants [88,91]. Transformants were selected with 100 and 50 mg/L kanamycin, respectively. More than fifteen independent transgenic lines (T1 generation) were obtained and genotyped by PCR. Three of them were randomly selected for propagation. The homozygous lines (T4 generation) were used in this study. For cloning of BnRH6-GFP, full-length CDS sequence of *BnRH6* without stop codon was fused into pCAMBIA1305 backbone through *Xba*/I*Bam*H I digestion and T4 ligation. 

### 4.3. Analysis of Transcripts by RT-PCR and qRT-PCR

Total RNA was extracted with Trizol (Invitrogen). Total RNA was treated with DNase I (Transgen, Beijing, China). The first-strand cDNA was generated by reverse transcription with EasyScript First-Strand cDNA Synthesis SuperMix (Transgen, Beijing, China) by Oligo(dT) RTprimer (mRNAs) according to the manufacturer’s instructions. For regular PCR reactions, PCR products were fractioned in 1% agarose gels. For qRT-PCR, the amplification systems were processed in a 20 μL solution with 10 μL TransStart Green qPCR SuperMix (Transgen, Beijing, China), 4 ng template, and 0.8 μL (10 μmol) primers (Appendix A). The reactions were run at 95 °C for 10 min, followed by 40 cycles of denaturation at 95 °C for 10 s and annealing at 60 °C for 1 min in the 7500 Real-Time PCR System (Applied Biosystems) [16]. *Brassica Actin2* serves as an internal control. All primers used are listed in Appendix A (Appendix A). Primers used for genotyping were designed on http://signal.salk.edu/tdnaprimers.2.html (accessed on 14 December 2022). *AtRH6*-qRT primers were synthesized according to [25]. *AtActin2* primers were synthesized according to [19]. *BnActin2* primers were synthesized according to [92]. Other primers were designed with Primer Premier6. 

### 4.4. Subcellular Localization Analysis

The pCAMBIA1305 (negative control), pCAMBIA1305-BnRH6-GFP, OsMDADS3-RFP (nucleus marker), and AtDCP1-RFP (P-body marker) were constructed and transformed into Agrobacterium tumefaciens strain GV3101. Agrobacterium tumefaciens containing different vectors were cultivated overnight and then suspended in the inoculation buffer (150 mM acetosyringone, 10 mM MgCl_2_, and 10 mM MES, pH 5.7) when the value of OD_600_ reached 1.0. Combinations of A. tumefaciens with different constructs were injected into four-week-old tobacco (Nicotiana benthamiana) leaves with a needleless syringe [90]. After three-day agroinfiltration, transiently expressed leaves were photographed with a confocal laser scanning microscope (LSM780, Zeiss, Oberkochen, Germany). 

### 4.5. Determination of Salt Stress Responses

For germination and cotyledon greening analysis, sterilized WT and *BnOE Brassica* seeds were placed on ½ MS medium containing 0, 100, 125, and 150 mM NaCl [93,94]. The germination proportion (emerged radicle) and cotyledon greening (expanded cotyledon turning green) were recorded from the 1st day (the day exposed to light) to the 7th day. Images were photographed on the 7th day. Sterilized WT and *AtOE Arabidopsis* seeds were placed on ½ MS medium containing 0, 50, 75, and 100 mM NaCl [95]. The germination proportion (emerged radicle) and cotyledon greening (expanded cotyledon turning green) were recorded and photographed on the 4th day and 7th day, respectively [96]. 

For root/shoot length and dry weight, ten-day-old *Brassica* plantlets were treated with ½ Hoagland solution containing 0, 100, 150, and 200 mM NaCl for 14 days. Sterilized WT and *AtOE Arabidopsis* were grown on ½ MS medium with 0, 50, 75, and 100 mM NaCl for 12 days. Sterilized WT and *rh6-1 Arabidopsis* seeds were placed on ½ MS medium containing 0, 45, and 70 mM NaCl for 14 days. The plates were vertically placed in a conditioned chamber. The root and shoot length were measured with a ruler, and the fresh mass and dry mass were weighted as described previously. [39,88,89,97]. 

### 4.6. Determination of Physiological Responses

For measuring seed germination, the surface-sterilized seeds of *Brassica* were placed on the solid 1/2 MS medium with the gradient NaCl concentrations at 0, 100, 125, and 150 mM. The germinated seeds (the seeds with emerged radicles) and the seeds with green cotyledons were counted every day from the 1st day to the 7th day. Surface-sterilized *Arabidopsis* seeds were sowed on solid 1/2 MS medium with the gradient NaCl concentrations at 0, 50, 75, and 100 mM. The germinated seeds (the seeds with emerged radicles) and the seeds with green cotyledons were counted on 4th day and 7th day, respectively. The calculation of seed germination ratio and cotyledon greening rate refers to [96]. 

For chlorophyll and MDA concentration measurements, *Brassica* seeds were hydroponically grown in ½ Hoagland solution for ten days and then grown in ½ Hoagland solution supplied with NaCl gradient from 0 to 100, 150, and 200 mM for 14 days. *Arabidopsis* seeds were potted in mixed soil (vermiculite and flower nutrient soil, 3:1) and placed in the chambers. Three-week-old *Arabidopsis* plants were irrigated with 0 and 200 mM NaCl solution for 12 days [98,99]. Total chlorophyll from *Brassica* or *Arabidopsis* was extracted with 80% (*v/v*) acetone and soaked for 36 h at room temperature in the dark until the leaves fully faded. The absorbance at 663 nm and 645 nm was recorded by a spectrophotometer [100]. The malondialdehyde (MDA) concentration was determined using a Lipid Peroxidation MDA Assay Kit (S0131, Beyotime, Shanghai, China) according to manufacturers’ instructions. 

### 4.7. Determination of Na^+^ and K^+^ Concentrations

Plant shoot and root tissues were separately harvested after NaCl treatment (the same as materials for chlorophyll and MDA measurements). The harvested tissues were immersed in 4 mM CaSO_4_ solution for 5 min. After being washed with deionized water, the samples were exsiccated in an oven at 72 °C for 72 h. The samples were digested with nitric acid, and the Na^+^ and K^+^ concentrations were determined by ICP-MS (PerkinElmer, Waltham, MA, USA) as previously described [39].

### 4.8. Construction of Salt Treated cDNA Libraries of Arabidopsis rh6-1 and RNA-Sequencing

*Arabidopsis* seeds of *rh6-1* and wild-type (WT, Col-0) were sowed on solid ½ MS medium for 14 days before being transferred to 1/2 Hoagland solution. The plants were then subjected to 200 mM NaCl treatment (+Na) with a 0 mM NaCl treatment (−Na) as a control (CK) for 6 h. 

Four groups of samples were harvested and total RNA was isolated. The samples were prepared as RNA-seq libraries in terms of WT(−Na), WT(+Na), *rh6-1*(−Na), and *rh6-1*(+Na). For each library, three biological samples with a total of 12 samples were prepared for RNA sequencing according to the previous method [101]. The Illumina RNA sequencing platform (HiSeq 2500) was applied. Data analyses were performed as described previously [89]. 

### 4.9. Statistical Analysis

The study was set up in biological triplicate. Each result shown in the figures was the mean of three replicated treatments, and each treatment contained at least 15–18 seedlings. 

With regard to the assessment of survival rates, 30 plants were employed. The significant difference between treatments was assessed through analyses of variance post hoc test (ANOVA, Tukey’s test). The values of each assay, followed by an asterisk, are significantly different at *p* < 0.05. The data were analyzed using the statistical software package SPSS 22.0.

## 5. Conclusions

This study functionally characterized a DEAD-box RNA helicase family gene *BnRH6* from *Brassica*. *BnRH6* can be transcriptionally induced by salt stress. Overexpression of *BnOE* revealed plant hypersensitivity to salinity stress, in terms of reduced germination rate, cotyledon greening, root and shoot elongation, chlorophyll concentration, and increased MDA concentrations in plants. These results were well-confirmed by similar studies with transgenic *Arabidopsis* (*AtOE*s) expressing *BnRH6*. Furthermore, both the *BnOE* and *AtOE* lines accumulated more Na^+^ than the wild-type. RNA-seq analysis revealed that a subset of downstream salt-tolerant genes was regulated by RH6, indicating that RH6 negatively regulates salt stress response by adjusting the downstream salt-tolerant genes. However, the detailed molecular mechanisms for how *BnRH6* regulates its downstream genes or proteins remain to be investigated. 

## Figures and Tables

**Figure 1 ijms-24-00002-f001:**
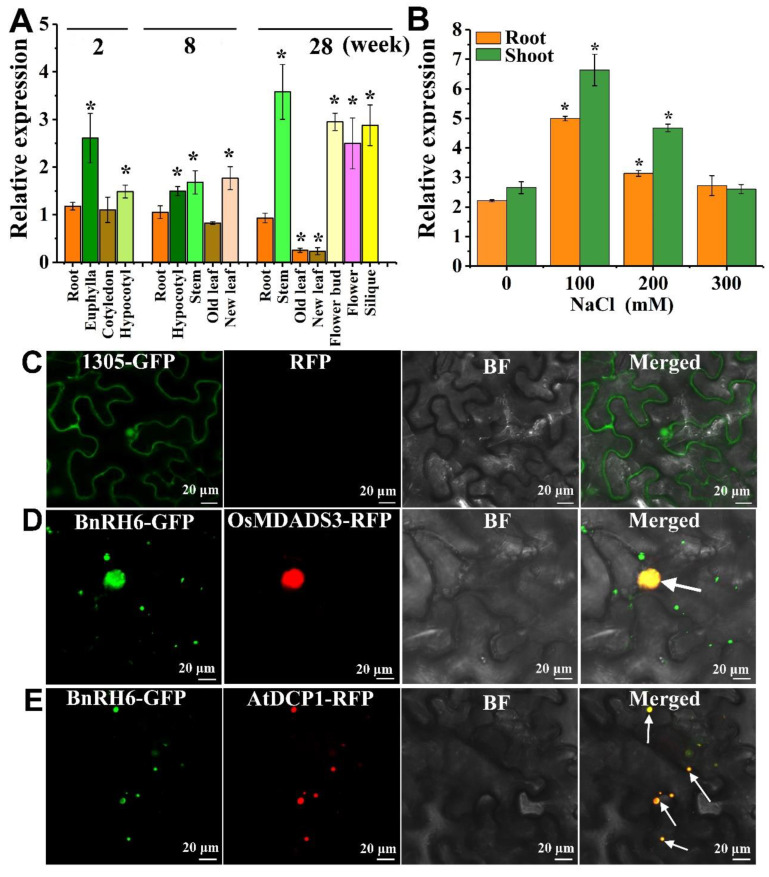
Analysis of BnRH6 transcripts and subcellular localization in *Brassica*. (**A**) Transcript levels of BnRH6 at different tissues or organs of *Brassica* across the life circle were determined by qRT-PCR. (**B**) *BnRH6* was transcriptionally induced under salt stress. Two-week-old seedlings were treated with 0, 100, 200, and 300 mM NaCl for 6 h. Vertical bars represent mean values ± SD (standard deviation) (*, *p* < 0.05, Student’s *t*-test). (**C**–**E**): the subcellular localization assay was conducted with tobacco leaves. (**C**): BnRH6-GFP fluorescence only. (**D**): BnRH6-GFP fusion was co-expressed with the nuclear marker (OSMDADS3-RFP). (**E**): BnRH6-GFP fusion was co-expressed with P-body marker (DCP1-RFP). The arrows pointed to the merged fluorescent signals. BF: bright field. Scale bar = 20 μm.

**Figure 2 ijms-24-00002-f002:**
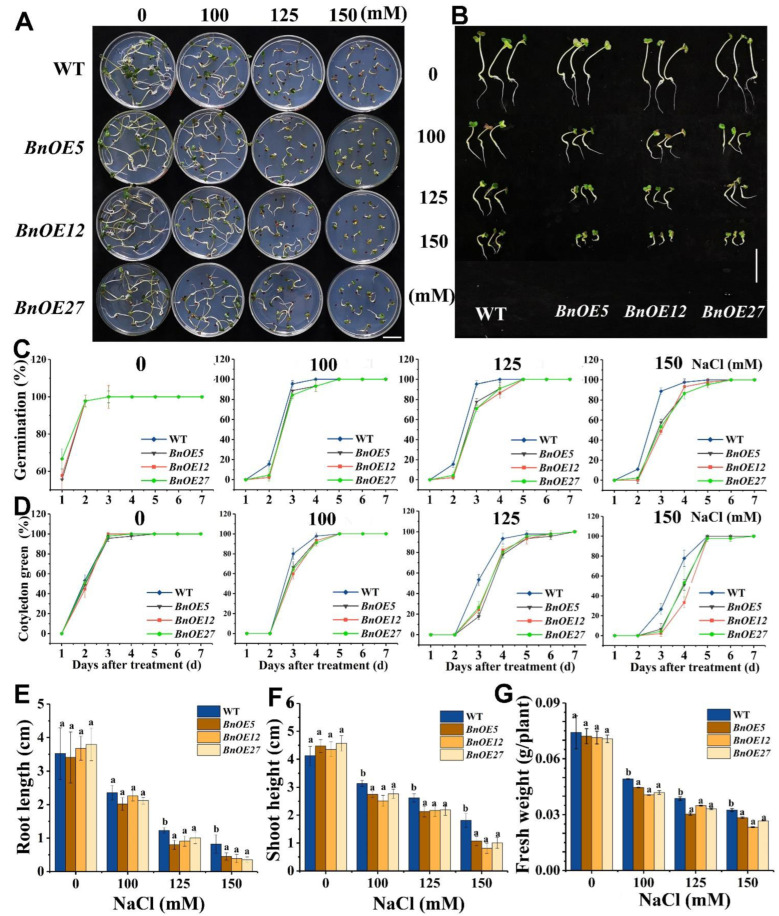
Seed germination and cotyledon greening of transgenic *Brassica* overexpressing *BnRH6* (*BnOE5*, *BnOE12* and *BnOE27*) lines and wild-type (westar) under salt stress. Seeds absorbing enough water were placed on the 1/2 MS medium with 0, 100, 125, and 150 mM NaCl for 7 days. (**A**,**B**): Phenotypes of *BnOE* lines. Scale bar = 3 cm. (**C**,**D**): Germination and cotyledon greening rates of *BnOE* lines. (**E**–**G**): Root length, shoot height, and fresh weight of *BnOE* lines. WT: wild-type. *BnOEs*: transgenic *Brassica* overexpressing *BnRH6.* Vertical bars represent mean values ± SD (standard deviation). Different letters indicate significant differences between three independent experiments (analysis of variance; *p* < 0.05).

**Figure 3 ijms-24-00002-f003:**
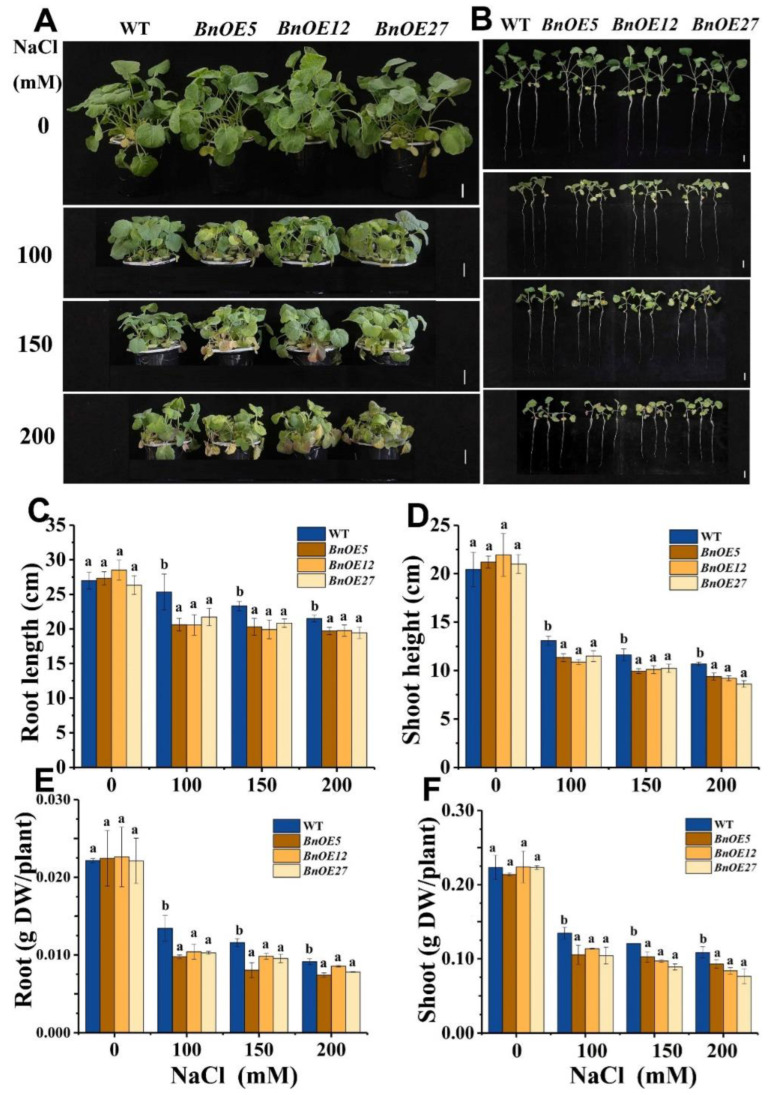
*BnRH6* overexpression reduces *Brassica* plant growth and dry weight under NaCl treatment. (**A**,**B**): Phenotypes of the transgenic BnRH6-overexpression *Brassica* (*BnOE*) lines. Scale bars = 3 cm. (**C**,**D**): Root length and shoot height of *BnOE* lines. (**E**,**F**): Root and shoot dry weight of *BnOE* lines. Ten-day-old wild-type and *BnOE* plants were treated with NaCl gradient from 75 to 100, 150, and 200 mM. After 14 days, the root length, shoot length, and dry weight were measured. DW: dry weight. Vertical bars represent mean values ± SD (n = 30). Different letters indicate significant differences between three independent experiments (analysis of variance; *p* < 0.05).

**Figure 4 ijms-24-00002-f004:**
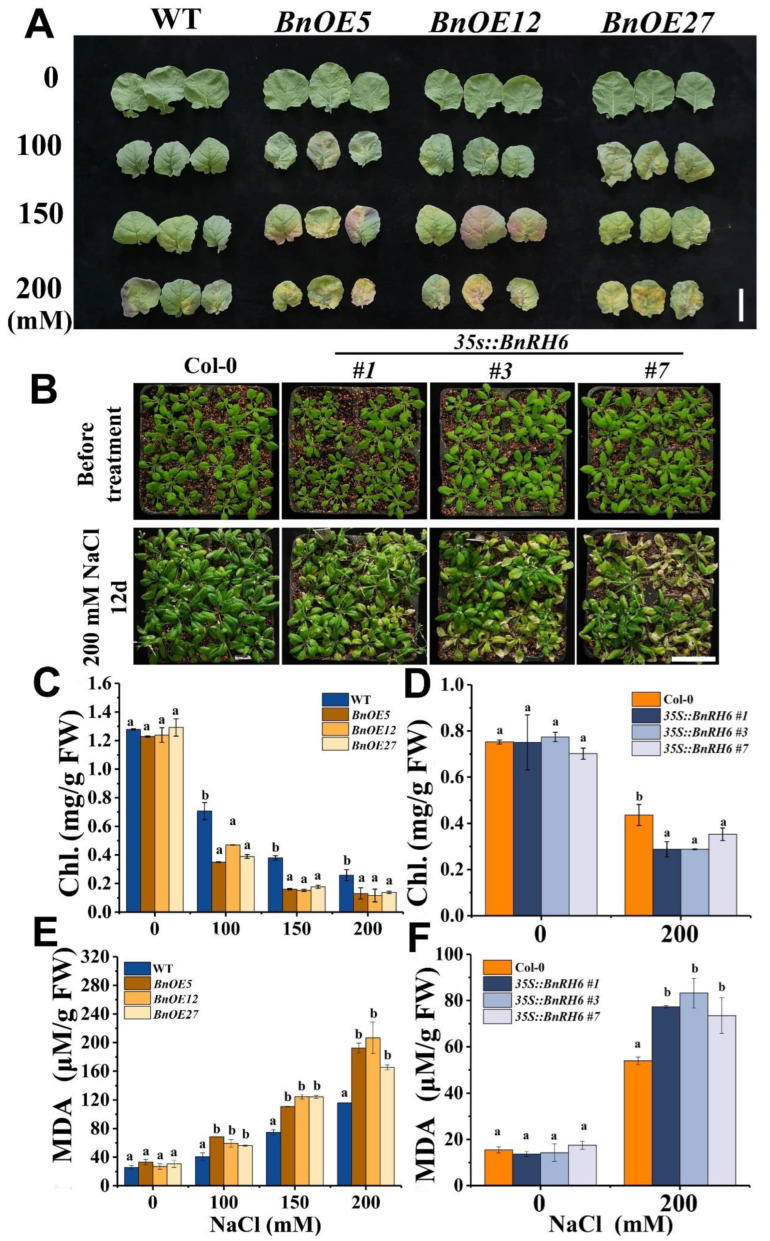
Effects of NaCl on the concentrations of chlorophyll and MDA in the transgenic BnRH6-overexpression *Brassica* (*BnOE*s) and *Arabidopsis* (35S:BnRH6s or *AtOE*). For *Brassica*, 10-day-old *BnOE* and wild-type plants were hydroponically grown and treated with NaCl at 0 to 100, 150, and 200 mM for 14 days. Regarding *Arabidopsis*, plants were potted in soil in conditioned chambers. Three-week-old *AtOE* and wild-type plants were treated with NaCl at 0 to 75, 100, 150, and 200 mM for 14 days. (**A**): Phenotypes of *Brassica* leaves under salinity stress. (**B**): Phenotypes of *AtOE* plants following treatment with 200 mM NaCl for 12 days. (**C**): Chlorophyll concentrations in *BnOE* plants. (**D**): Chlorophyll concentrations in *AtOE* plants. (**E**): MDA concentrations in *BnOE* plants. (**F**): Chlorophyll concentrations in *AtOE* plants. Chl: Chlorophyll. MDA: malondialdehyde. FW: fresh weight. Vertical bars represent mean values ± SD (n = 30). Different letters indicate significant differences between three independent experiments (analysis of variance; *p* < 0.05). Sacle bar = 3 cm.

**Figure 5 ijms-24-00002-f005:**
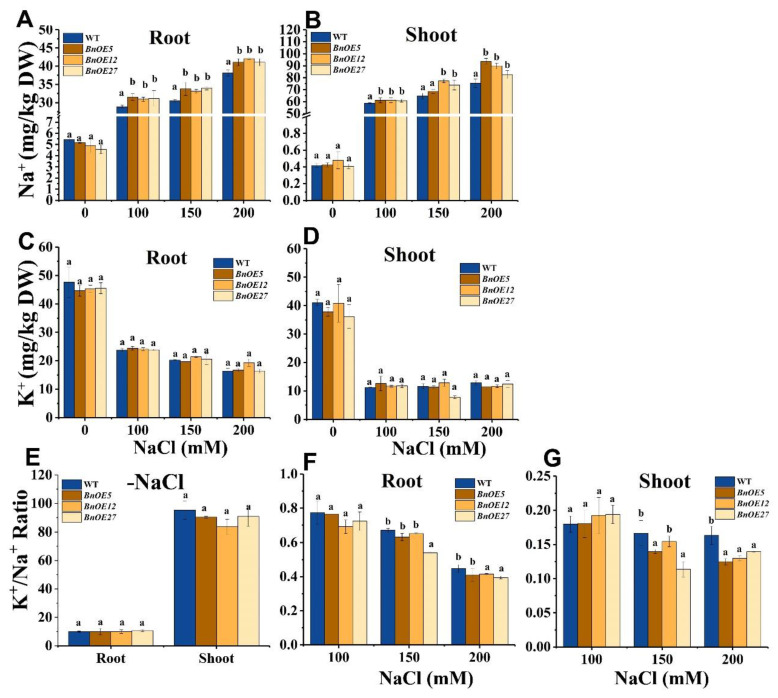
Analysis of the concentration of Na^+^ and K^+^ and K^+^/Na^+^ ratio of root and shoot in WT and *BnOE*. (**A**,**B**): Na^+^ concentration of root and shoot in WT and *BnOE*. (**C**,**D**): K^+^ concentration of root and shoot in WT and *BnOE*. (**E**–**G**): K^+^/Na^+^ ratio of root and shoot in WT and *BnOE*. Ten-day-old wild-type and *BnOE* plants were treated with NaCl gradient from 0 to 75, 100, 150, and 200 mM or without for 14 days. The root and shoot were harvested and dried at 80 °C for about 24 h. The concentration of Na^+^ and K^+^ of all samples was analyzed using ICP-AES. DW: dry weight. Vertical bars represent mean values ± SD (n = 30 seedlings). Different letters indicate significant differences between three independent experiments (analysis of variance; *p* < 0.05).

**Figure 6 ijms-24-00002-f006:**
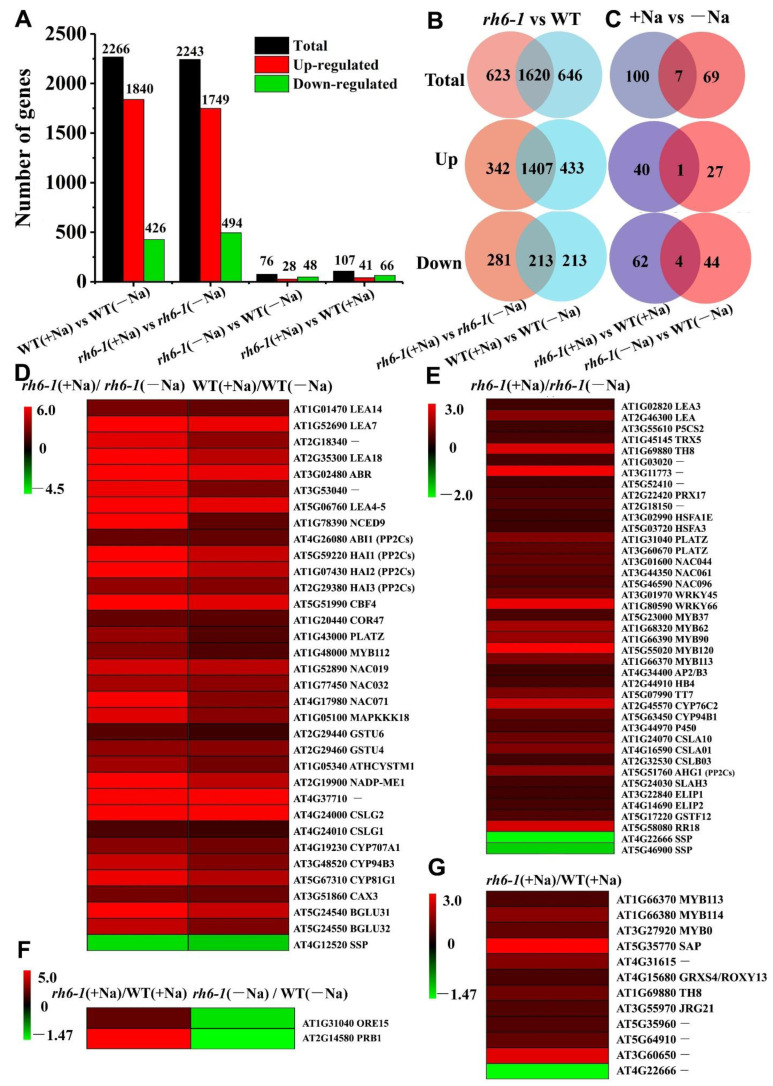
Differential genes in expression of WT and *rh6-1* under salt stress determined by RNA-seq. (**A**) The number of differential genes in expression in four datasets. (**B**) Venn diagram showing total, up- and downregulated genes in Col-0 under 200 mM NaCl stress (+NaCl) relative to normal condition (−NaCl), and in *rh6-1* mutant seedlings under the same stress relative to normal condition. (**C**) Venn diagram showing total, up- and downregulated genes in *rh6-1* seedlings relative to Col-0 grown under (+NaCl) and (−NaCl). (**D**): Heat map of 34 co-expressed genes in WT(+Na) vs. WT(−Na) and *rh6-1*(+Na) vs. *rh6*-*1*(-Na) datasets. (**E**) Heat map of 41 unique genes of *rh6-1*(+Na) vs. *rh6*-*1*(−Na) dataset. (**F**): Heat map of 2 co-expressed genes in *rh6*-*1*(−Na) vs. WT(−Na) and *rh6*-*1*(+Na) vs. WT(+Na) dataset. (**G**): Heat map of 12 unique genes of *rh6*-*1*(+Na) vs. WT(+Na) dataset.

**Figure 7 ijms-24-00002-f007:**
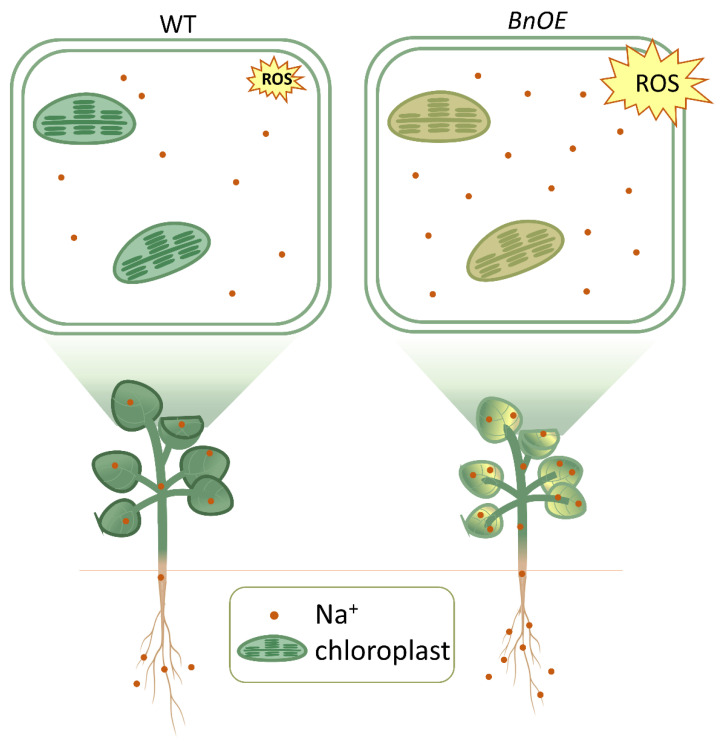
Overview of salt stress responses of WT and BnOE *Brassica* plants. ROS: reactive oxygen species.

## Data Availability

Not applicable.

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
