# Peer review of "Identification of a DEAD-box RNA Helicase BnRH6 Reveals Its Involvement in Salt Stress Response in Rapeseed (Brassica napus)"

_ijms, 2022, doi:10.3390/ijms24010002_

Round 1
Reviewer 1 Report (Previous Reviewer 1)
Figure 1: please, mentioned that panels C- E is tobacco leaf.
Line 120: must be fluorescence.
Line 132: maker or marker?
Line 139: do you mean overexpressed?
Figure 4 B – scale bar
Figure 7: please, clarify which cell type you describe in the model and include epigenetic part.
Line531: “and an” ¿?
Line 520-527: plesae, clarify about sterility of the culture.
Line 534: 3% sucrose?? It is too much. Moreover, one have consider that in nature plants do not have sucrose and NaCl mainly affected photosynthesis and therefore growth/metabolism.
Line 535: “grown in a growth chamber” ¿?
How did you count germination? Plesae, clarify. Moreover, germination must be studied without cold treatments, what significantly change results.
Lines 610-635: plesae, clarify these sentences, remove some contradictions.
Author Response
Please see the attachment.

Reviewer 2 Report (Previous Reviewer 2)
The authors have addressed most of my (substantive) comments. On the formal side, however, the manuscript suffers from a non-standard way of writing, e.g. Latin names. Even Brassica or Arabidopsis is a Latin name that should be italicized. In dealing with comments, for example, they state that they have modified the Reference part (unification of journal names, capitalization and lowercase letters), but this is not the case in the new form of the manuscript, or I don't see these modifications. I have a fundamental comment about the appendices, where the authors have learned their lesson and have a self-explanatory legend here too, but with composite images, their size, or readability. An example is, for example, Figure S7 (C), where numerical values are not noticeable even at a magnification of 200%!
The methodology used a large number of primers for different types of analyses. Since there is no information in the methodology about the procedure for designing primers by the authors of the manuscript, I deduce that these are adopted primers. In this case, a reference should be given for individual primers and these should then be listed (this absence is a violation of publication ethics). If it is the result of the authors, it is necessary to supplement the relevant methodology. Without this, the manuscript cannot be recommended for publication!
Author Response
Please see the attachment.

Reviewer 3 Report (Previous Reviewer 3)
The authors have addressed my and other reviewers' concerns properly and the quality of the manuscript has improved significantly.
Author Response
Thank you very much for your approval!
Round 2
Reviewer 1 Report (Previous Reviewer 1)
Thank you for clarification and correction.
However, some points are required further explanation/corrctions.
„Figure 7: please, clarify which cell type you describe in the model and include epigenetic part. Answer: We have withdrawn our previous model and put a new one which was made exactly based on our research results as below. In our updated version, we have labeled the WT and BnOE clearly. Again, as BnRH6 is ubiquitously expressed (Fig. 1A in our manuscript), the cell does not refer to any specific type of cell“
Figure 1A did not demonstarte expression level per cell type, but demonstatde very low level in the leaf. The most importnat is cell type expression. Für example, spongy mesophyll and vascular cells play different role and chnages expression in one cell type may have a greta effcet, while expression in the other type does not have any effect. Please, discuss this point!
Line 534: 3% sucrose?? It is too much. Moreover, one have consider that in nature plants do not have sucrose and NaCl mainly affected photosynthesis and therefore growth/metabolism. Answer: Brassica were grown in ½ Hoagland solution with no sugar at all. Arabidopsis were grown on ½ MS solid medium. The sucrose concentration in ½ MS solid medium is 30g/L, because Arabidopsis need extra carbon resource for growth. This is a widely-used standard formula for Arabidopsis culture and also used in salt treatment, as high sucrose stimulates root growth and makes it easier to find mutants with inhibited root growth (Verslues, Agarwal et al. 2006, Wang, Kim et al. 2009, Gao, Song et al. 2017, Zhao, Feng et al. 2017).
Thank you for explanation. Please, „consult“ the literrature that medium you have used are very toxic for Arabidopsis and generated high stress itself. Itb is alos very inetersting that in your case 3% sucrose does not inhibited root griwth. In my cvase, after 20 years studying Arabidopsis stress response, I can not confirmed this point.
In my case 1% of sucrose can promoting plant gprowth in the case of low light intensity. In the normal light no differences have been observed.
Moreover, presence of sucrose in my case caused protection form stress (salt sztress have an effcet on photosyntrehsis and carbohydtae supply in the case of absent external one). And this, in turn, prevent root and plant growth inhibition.
Please, clarify all these points.
How did you count germination? Plesae, clarify. Moreover, germination must be studied without cold treatments, what significantly change results. Answer: As we explained in the earlier response letter, we consider the seeds with emerged radicles as germinated seeds. We count them on the indicated days. A cold treatment at 4 °C for 2-3 days will improve the rate and synchrony of germination. In another word, this short[1]term cold treatment gets the seeds ready to germinate. This is a widely used treatment for Arabidopsis and other seeds (Rivero, Scholl et al. 2014).
Scientificaly, seeds germination is dormancy broken and phenotypicaly must be count as testa rupture in Arabidopsis. Once testa rupture happens, seeds can not became dormant anymore, cant dry without lost of viability. Probably you, as some other authros, measure post-germinated seedling growth. Seed germination is chormatin „activation“ , while seedlings griwth is a complicated process require cell divsion, differentiation, elongation etc..
Please, mention these points in discussion.
My best regards!
Author Response
Please see the attachment

Reviewer 2 Report (Previous Reviewer 2)
The authors accepted all my comments. The manuscript can be accepted for publication.
Author Response
Thank you very much for your approval.
This manuscript is a resubmission of an earlier submission. The following is a list of the peer review reports and author responses from that submission.
Round 1
Reviewer 1 Report
It is quite inetersting story, but a lot of clarifications are required.
Line 19: This study investigated the function and regulatory mechanism for BnRH6. – please, edit.
BnRH6. 19 BnRH6 ?? why you used both normal and italic?
Line 22: B. napus and Arabidopsis ? Please, make uniform. Either Brassica and Arabidopsis or B.napus and A.thaliana.
Line 23-24: : „oxidative stress, and over-accumulation of Na ions with the K+ /Na+ ratio being decreased by 18.3%-28.6%.“ are not a phenotype.
Figure 1.
A – please, clarify differences between meristemic zone oft he root and mature zone, different cell type in the stem etc. How this can be different.
B – how can you explin down regulation of gene expression under 300 mM NaCl? How did you do treatment? Have you change 20 parameters or only one?
For such treatments you must to change only one parametr, ea. Add NaCl to existing solution, not replacement solution. By replacing solution you add all new chemicals as Ca, Mg etc.
Lines 115- 127: here you show expression only in one cell type with dominat nuclei and low expression oft he other egene. Form this picture you can not generized expression patterns for all cell type. Do you have the same expression of gene in dividing cell of the root (what is the main source of cell during root growth)? These cells have a different structure and different organelles. Please, discuss this point. This is very importnat because you have used cell with extremely low gene expression level (figure 1A, old leaf) for analysis.
Fig 2: A, B – scale bar are missing.
It is not clear what do you mean as seeds germination. Seeds germination is specific realisation from dormancy. What can be count as testa rupture (after 24 hours upon soaking). Here graphs C are unclear: how did you measure? And, in summary you wrote about reducing in seedlings establishment (not seeds germination) what is not fit with figure...
Figure 3, line 177: scale bar. Not white bar!
Please, expian how do you measure dry weight.
Figure 4: scale bar are missing.
Figure 4B: is not correct. You have to show control after 12 d, not before treatments.
Figure 7:
Please, specify whioch cell type do you mean? Cell in cell cycle of mesophyll cell? Etc?
What is growth regulatiory gene?
GST is glutathione transfrerase, it is not ROS scavenging one.
Line 492: nutrient solution??
Line 493: „Plants growing solid medium were sown in mixed soil“ ?? what do you mean?_
Line 535: transit or transient?
Line 538: „Physiological phenotype assays“?? Why not to write phenotyping?
Line 542: what do you eman as „seed germination rate and cotyledon greening“ ? Testat ruputure and spectrum of the cotyledon? Have you count spectrum pre whole area or another way?
Seeds germination end is testa rupture after which seeds can not come back to dormancy and died if drying. Plesae, clarify points.
It is alos very suprisingly that you used two different medium for assays: toxic MS (Van Delden, S. H., Nazarideljou, M. J., & Marcelis, L. F. (2020). Nutrient solutions for Arabidopsis thaliana: a study on nutrient solution composition in hydroponics systems. Plant methods, 16(1), 1-14.) with high nutional stress, including „turgor“ stress and Hoagland without turgor stress and nutrional stress.
In addition, NaCl may have two effects: osmotoc stress as Na and turgor stress as chloride. Please, mentio this in discussion.
Lines 544- 549: these two test can not be compared becuase you have sued different conditions. You need to use similar conditions, for example, Hoagland medium. Moreover, you did not menti how much sucrose contain your mediun in vitro. It is importnat in your case since Na has a strong effcet on photosyntehsis. An optimal design is to used in both case hydroponic with 1/2 Hg medium (witoot sucrose, of course).
Line 555: „from rape or Arabidopsis“ – please, use the same terminology! Arabidopsis should not be italic, in this case.
Reviewer 2 Report
According to the title and abstract, the manuscript looks very interesting and up-to-date. However, when I read the PDF version of the full text of the manuscript, I was disappointed, because the manuscript contains fundamental flaws that do not allow us to write anything other than a negative opinion (reject).
The authors do not respect the standards, for example, when writing botanical names in italics, the same is the case with genes, or of allelic variants, writing restriction enzymes, etc. Is it RNP (line 46 right?), if so, then the abbreviation must be explained. I consider the fact that the authors refer in the results section, even the methodology, to appendices that are not part of the manuscript. Even the numbering does not correspond to the fact that the order is implemented according to the order of references in the text. This fundamental absence does not allow verifying the facts that the authors describe in the methodology, results or are even discussed. Similar flaws are evident in some images where the legend is complete, i.e. including the description of line segments, where in Fig. 2 it is written that they indicate the standard deviation of mean. The question is whether this is also the case with other data (figures) that are presented in the same way. If abbreviations are used in the pictures, they must always be explained, but the self-explanatory content of the picture has been filled. References appear in some parts of the results (i.e. line 102, 125, 126) and the sentence formulation clearly indicates a discussion. This should be elsewhere in the manuscript, i.e. in discussion. The level of discussion is very low and very often at a general level, using a reference to multiple citations in one place (example is subsection 3.1.1. or 3.1.2). The discussion should be critical and reflect the authors' reflections on reasons for agreement or differences with other authors. The methodology contains facts that are not supported by citations, even if it concerns, for example, the composition of media or nutrient solutions. The material used is not always clearly specified, e.g. the specification of the tobacco genotype. There is a lack of justification for individual treatment options. How and why did the authors determine the NaCl concentrations used in the study? In the References section, there are errors in the names of journals, where uppercase and lowercase letters are very often not respected. The manuscript would also deserve a slight proofreading of the English language to remove minor typos and less than ideal word formulations (connections).
Reviewer 3 Report
Dear Editor/Authors:
I have gone through the assigned manuscript entitled with" Identification of a DEAD-box RNA helicase BnRH6 reveals its involvement in salt stress response in rapeseed (Brassica napus)" which is submitted for potential consideration to publish in the journal (IJMS). In this study the authors reported that BnRH6 (a DEAD-box RNA helicase) as a negative regulator plays a primary role in B. napus adaptation to the abiotic stress (salt stress). The manuscript contains numerous flaws in structure, methods, and write up and presentation, which make it difficult to read in its present form. The literature review offers a useful overview of current research and the resulting bibliography provides a very useful resource for mustard molecular breeders. I have no hesitation in recommending that it be accepted for publication after revision of material and method section, typos and other minor details have been attended to.
1. In abstract, methodology is not clear, most of the proportion is reflection result section it should reflect significant findings justifying your research hypothesis. State clearly the principal conclusion.
2. In material and method section, at several places there is lacking of coherence and confusing statements, as I cannot understand what the authors are trying to communicate, please rephrase this section meaningfully (for example in lines 493-494, 511-512, 542-543).
3. Overall the discussion section can be improved significantly. A more critical discussion is warranted involving, what are the implications of key findings and how the research findings could be helpful for molecular mustard breeder.
4. I would have wished to see conclusion in a separate section, having the take home message for the molecular breeders